# LEARNING LOSS LANDSCAPES IN PREFERENCE OPTIMIZATION

## ABSTRACT

We present a framework to discover preference optimization algorithms specialized to particular scenarios, in a theoretically sound and computationally efficient setting. We start by designing a novel family of PO algorithms based on mirror descent, which we call Mirror Preference Optimization (MPO). MPO recovers existing methods like Direct Preference Optimization (DPO) and Odds-Ratio Preference Optimization (ORPO) for specific choices of the mirror map. Given specific properties of preference datasets, such as mixed-quality or noisy data, we show that we can efficiently search the MPO class to find specialized algorithms that outperform current baselines. Namely, we leverage evolutionary strategies and preference datasets generated on MuJoCo environments to systematically evaluate and optimize MPO algorithms on hand-crafted scenarios. We demonstrate the resulting PO algorithms successfully transfer to a Large Language Model (LLM) alignment task, where they demonstrate superior robustness in handling mixed-quality datasets.

## 1 INTRODUCTION

Learning from human feedback is a paradigm that enables the alignment of complex agents to human preferences, and has been successfully applied to Large Language Models (Team et al., 2023; Achiam et al., 2023). In particular, fine-tuning pretrained LLMs with human preferences has become a popular strategy to adapt them to specific tasks and to improve their safety and helpfulness.

Most LLM alignment pipelines begin with a supervised fine-tuning (SFT) step, which involves supervised next-token prediction on a dataset of high-quality responses and leads to a reference policy. The reference policy is further optimized using the human preference data, typically through either Reinforcement Learning from Human Feedback (RLHF) (Christiano et al., 2017) or Direct Preference Optimization (DPO) (Rafailov et al., 2024), or one of their several variants. RLHF consists of learning a reward model consistent with human preferences and then using Reinforcement Learning (RL) techniques such as REINFORCE (Sutton et al., 1999) and Proximal Policy Optimisation (PPO) (Schulman et al., 2017) to maximize the total expected reward. In contrast, DPO and its variations, e.g. odds ratio preference optimization (ORPO) (Hong et al., 2024), bypass explicit reward models entirely and optimize directly on preference data, implicitly learning the reward.

While RL-based methods offer stronger theoretical guarantees and often lead to higher performance (Song et al., 2024; Xu et al., 2024a), offline approaches such as DPO have gained traction due to their simplicity and the ability to leverage preexisting high-quality datasets. In contrast to PPO, where data collection and labeling are performed iteratively after each update, DPO and its modifications allow for more efficient training, avoiding the high computational costs, need of additional sample labelling and complexity of RL methods. Specifically, PPO requires careful parameter tuning (Yuan et al., 2023) and involves simultaneous training of multiple models (the reward model, language model, and critic), which can be prohibitive in most hardware setups.

The performance of offline algorithms such as DPO, particularly on noisy or low-quality datasets, has been a subject of debate (Chowdhury et al., 2024), with limited empirical results available. In this work, we provide a comprehensive analysis of PO algorithms, examining their behavior on automatically generated preference datasets. We perform this analysis in MuJoCo environments (Todorov et al., 2012), where the underlying reward structure is well defined and offers a clear performance metric to compare agents. In particular, we focus on ORPO, as we have found

that it largely outperforms DPO in all settings we have considered. Additionally, our findings indicate that ORPO exhibits distinct failure modes when applied to specific low-quality or noisy datasets. These failure modes are present in practical LLM applications, raising concerns about using mixed-quality datasets for PO.

We introduce a novel framework to find algorithms that are capable of dealing with these more difficult settings. We first define a novel class of Preference Optimization (PO) algorithms based on mirror descent (Nemirovski & Yudin, 1983), which generalizes DPO and ORPO for particular choices of the mirror map. Our framework consists in searching the MPO class using evolutionary strategies (ES) and hand-crafted preference datasets generated on MuJoCo environments, with the goal of discovering new and better performing algorithms. We outline our contributions in detail below.

1. We perform a systematic analysis on the performance of ORPO on automatically generated preference datasets with varying levels of data quality, noise levels, initial policy, and judge temperature.

2. We introduce a novel family of offline PO algorithms using mirror descent, named Mirror Preference Optimization (MPO), which can be easily parameterized and explored via ES.

3. For each failure mode we identify, we find and describe an algorithm within our framework that outperforms ORPO. Additionally, we show that allowing losses to vary based on the percentage of training progress drastically boosts performance in some settings.

4. We demonstrate that PO algorithms discovered on MuJoCo can generalize to LLM tasks. In particular, for a highlighted failure setting, we show that our discovered algorithm improves performance w.r.t. the ORPO baseline. This finding showcases the efficacy of analyzing PO algorithms in simpler, less computationally expensive environments, as it is possible to obtain insights relevant for LLM applications.

5. Following our findings, we present a closed form expression for one of our discovered PO algorithms that can be implemented with one line of code.

## 2 PRELIMINARIES

Let $\mathcal{M} = (\mathcal{S}, \mathcal{A}, P, r, T, \mu)$ denote an episodic Markov Decision Process, where $\mathcal{S}$ and $\mathcal{A}$ are respectively the state and action spaces, $P(s' \mid s, a)$ is the transition probability from state $s$ to $s'$ when taking action $a$, $r(s, a) \in [0, 1]$ is the reward function, $T$ is the maximum episode length, and $\mu$ is a starting state distribution. A policy $\pi \in (\Delta(\mathcal{A}))^{\mathcal{S}}$, where $\Delta(\mathcal{A})$ is the probability simplex over $\mathcal{A}$, represents the behavior of an agent on an MDP, whereby at state $s \in \mathcal{S}$ the agents takes actions according to the probability distribution $\pi(\cdot \mid s)$. Let $\tau = \{(s_t, a_t)\}_{t=0}^{T-1}$ denote a trajectory of length $T$ and, with a slight overload of notation, let $\pi(\tau) = \prod_{t=0}^{T-1} \pi(a_t|s_t)$ and $r(\tau) = \sum_{t=0}^{T-1} r(s_t, a_t)$. Lastly, let $\pi(\cdot|\tau)$ be the product distribution of $\pi(\cdot|s_0), \ldots, \pi(\cdot|s_{N-1})$.

Let $\mathcal{D} = \{(s_0^i, \tau_w^i, \tau_l^i)_{i=1}^N\}$ be a preference dataset, where each tuple $(s_0, \tau_w, \tau_l)$ consists of a starting state $s_0$ and two trajectories with starting state $s_0$. Each pair of trajectories is ranked by a judge, who determines a chosen trajectory $\tau_w$ ("win") and a rejected trajectory $\tau_l$ ("lose"), based on the cumulative rewards $r(\tau_w)$ and $r(\tau_l)$. We assume the judge ranks trajectories according to the Bradley-Terry model (Bradley & Terry, 1952), whereby the probability of choosing $\tau_w$ over $\tau_l$ is defined as

$$\mathbb{P}(\tau_w \succ \tau_l) = \frac{\exp(r(\tau_w)/\eta)}{\exp(r(\tau_w)/\eta) + \exp(r(\tau_l)/\eta)} = \sigma((r(\tau_w) - r(\tau_l))/\eta), \quad (1)$$

where $\sigma$ is the sigmoid function and $\eta$ is a temperature parameter. Our objective is to exploit the dataset $\mathcal{D}$ to find a policy $\pi^\star$ that maximizes the expected cumulative reward of an episode, that is

$$\pi^\star \in \underset{\pi}{\operatorname{argmax}} \, \mathbb{E}_{\tau \sim (\mu, \pi, P)} r(\tau) := \underset{\pi}{\operatorname{argmax}} \, \mathbb{E}_{s_0 \sim \mu, a_t \sim \pi(\cdot|s_t), s_{t+1} \sim P(\cdot|s_t, a_t)} \sum_{t=0}^{T-1} r(s_t, a_t). \quad (2)$$

In this work we consider an offline setting during training, where we do not have access to either the transition probability $P$, the reward function $r$, or the MDP $\mathcal{M}$. During evaluation, we calculate cumulative rewards online.

## 2.1 Preference Optimization

There are several methods in the literature to optimize the objective in (2) using a preference dataset $\mathcal{D}$. We provide here a short review of the main pipelines (Ouyang et al., 2022). Two common independent preliminary steps are reward modelling (RM) and supervised fine tuning (SFT). RM aims to obtain an estimate of the true reward function, and is usually framed as a maximum likelihood estimation problem for a Bradley-Terry preference model, i.e. find

$$\widehat{r} \in \underset{r_\theta}{\arg\max} \log \sigma(r_\theta(\tau_w) - r_\theta(\tau_l)/\eta), \tag{3}$$

for a parametrized reward class $\{r_\theta : \theta \in \Theta\}$ and for $\eta \geq 0$. SFT is an initial alignment phase, where the starting policy $\pi_0$ is trained to imitate high-quality demonstration data. In particular, the starting policy $\pi_0$ is updated to minimize the cross-entropy loss $\ell(\pi, (s_0, \tau_w, \tau_l)) = -\log(\pi(\tau_w))$, typically doing one epoch over the dataset. We call *reference policy* $\pi_{\text{ref}}$ the policy obtained at the end of this procedure. The new objective we want to optimize is

$$\pi^\star \in \underset{\pi}{\arg\max}\, \mathbb{E}_{s_0 \sim \mathcal{D}, \tau \sim (\pi, P)} \left[ \sum_{t=0}^{T-1} \mathbb{E}_{a \sim \pi(\cdot|s_t)}[r(s_t, a)] - \beta D_{\text{KL}}(\pi(\cdot|\tau), \pi_{\text{ref}}(\cdot|\tau)) \right], \tag{4}$$

where the $D_{\text{KL}}$ represents the KL-divergence and is introduced to prevent the policy from moving too far away from the dataset distribution. The expressions in (3) and (4) can be optimized sequentially, first obtaining a reward estimate in (3) and then optimizing (4) with PPO using the reward estimate in place of the true reward. Alternatively, the optimization problems in (3) and (4) can be solved implicitly using DPO or ORPO. We proceed to discuss each of these methods.

**PPO** PPO is recognized as one of the preferred methods for optimizing (4) when the necessary computing resources are available, as demonstrated by its success in training state of the art models like GPT-4 (Achiam et al., 2023) and Claude (Antropic, 2023). However, it presents a complex pipeline where one needs to effectively train a reward model, perform SFT and then optimize (4), where each phase has a different set of hyper-parameters. Additionally, storing the reward model, the reference agent and the current agent in memory is impractical in most setups and often requires sacrificing other aspects, such as batch-size. Besides it computational costs, PPO is known to be prone to reward overoptimization (Coste et al., 2024; Gao et al., 2023).

**DPO** Direct Preference Optimization (DPO) bypasses the need for an explicit reward model by using the agent itself to implicitly represent the reward model. It consists in optimizing the objective

$$\pi^\star \in \underset{\pi}{\arg\max}\, \mathbb{E}_{(s_0, \tau_w, \tau_l) \sim \mathcal{D}} \left[ \log \sigma \left( \beta \left( \log \frac{\pi(\tau_w)}{\pi_{\text{ref}}(\tau_w)} - \log \frac{\pi(\tau_l)}{\pi_{\text{ref}}(\tau_l)} \right) \right) \right], \tag{5}$$

which is obtained by plugging the theoretical solution of (4) in the maximum likelihood problem in (3). Refer to Appendix B for details. Thanks to its simplicity, DPO has been widely adopted to fine-tune LLMs as an alternative to PPO (Yuan et al., 2024; Jiang et al., 2024).

A known issue of DPO is that it pushes probability mass away from the preference dataset and to unseen responses (Xu et al., 2024b), which can cause the final policy to deviate significantly from the reference policy, even when the reference policy aligns well with human preferences. In contrast, PPO can leverage the generalization capabilities of the (learned) reward model to generate responses beyond the preference dataset distribution, while the KL-divergence penalty can provide additional regularization. To mitigate the risks described above, DPO is usually only applied for a few epochs.

**ORPO** ORPO further simplifies the training pipeline and addresses the distribution shift issue present in DPO. It merges the SFT and DPO steps into one, optimizing the unified objective

$$\pi^\star \in \underset{\pi}{\arg\max}\, \mathbb{E}_{(s_0, \tau_w, \tau_l) \sim \mathcal{D}} \left[ \underbrace{\log \pi(\tau_w)}_{\text{SFT}} + \lambda \underbrace{\log \sigma\left(\log\left(\text{odds}_\pi(\tau_w)\right) - \log\left(\text{odds}_\pi(\tau_l)\right)\right)}_{\text{preference optimization}} \right] \tag{6}$$

where $\text{odds}_\pi(\tau) = \pi(\tau)/(1 - \pi(\tau))$. ORPO gets rid of the need for a reference model by adding an SFT term to the preference optimization objective function, and uses this term to prevent the optimized policy from moving too far away from the dataset distribution. Additionally, the SFT term prevents pushing probability mass away from the preference dataset, addressing the distribution shift issue present in DPO.

## 2.2 MIRROR MAPS

We review the concept of mirror map, which will be needed when describing our methodology. For a convex set $\mathcal{X} \subseteq \mathbb{R}^{|\mathcal{A}|}$, a *mirror map* $h : \mathcal{X} \to \mathbb{R}$ is defined as a strictly convex, continuously differentiable and essentially smooth function[1] function that satisfies $\nabla h(\mathcal{X}) = \mathbb{R}^{|\mathcal{A}|}$. Essentially, a mirror map is a function whose gradient allows bijective mapping between the primal space $\mathcal{X}$ and the dual space $\mathbb{R}^{|\mathcal{A}|}$. The specific class of mirror maps that we are going to use is the $\omega$-potential mirror map class, to which most mirror maps considered in the literature belong.

**Definition 2.1** ($\omega$-potential mirror map Krichene et al. (2015)). For $u \in (-\infty, +\infty]$, $\omega \leq 0$, an *$\omega$-potential* is defined as an increasing $C^1$-diffeomorphism $\phi : (-\infty, u) \to (\omega, +\infty)$ such that

$$\lim_{x \to -\infty} \phi(x) = \omega, \ \lim_{x \to u} \phi(x) = +\infty, \ \int_0^1 \phi^{-1}(x)dx \leq \infty.$$

For any $\omega$-potential $\phi$, we define the associated mirror map $h_\phi$ as

$$h_\phi(\pi(\cdot|s)) = \sum_{a \in \mathcal{A}} \int_1^{\pi(a|s)} \phi^{-1}(x)dx.$$

When $\phi(x) = e^{x-1}$ we recover the negative entropy mirror map, while we recover the $\ell_2$-norm when $\phi(x) = 2x$ (refer to Appendix C). Mirror maps in this class are simple to implement in practice, where $\mathcal{A}$ is often large, as they can be parametrized by a scalar function instead of a multi-dimentional one. Additionally, the same $\omega$-potential $\phi$ can be used to generate mirror maps for different action spaces, allowing the insights obtained for one action space to easily generalize to others. An $\omega$-potential mirror map $h_\phi$ induces a *Bregman divergence* (Bregman, 1967), which is defined as

$$\mathcal{D}_{h_\phi}(\pi(\cdot|s), \pi'(\cdot|s)) := h_\phi(\pi(\cdot|s)) - h_\phi(\pi'(\cdot|s)) - \langle \nabla h_\phi(\pi'(\cdot|s)), \pi(\cdot|s) - \pi'(\cdot|s) \rangle,$$

where $\mathcal{D}_{h_\phi}(\pi(\cdot|s), \pi'(\cdot|s)) \geq 0$ for all $x, y \in \mathcal{Y}$. When $\phi(x) = e^{x-1}$, $\mathcal{D}_{h_\phi}$ is equivalent to the KL-divergence, while we recover the Euclidean distance when $\phi(x) = 2x$ (refer to Appendix C). When the Bregman divergence is employed as a regularization term in optimization problems, tuning the mirror map allows us to control the geometry of the updates of the parameters to be optimized, determining when to take large or small updates based on the current value of the parameters.

## 3 METHODOLOGY

We develop a new framework for preference optimization based on mirror maps, which generalizes DPO and ORPO. We start by replacing the KL-divergence penalty term in the objective in (4) with a more general Bregman divergence, that is, we aim to solve the problem

$$\pi^\star \in \underset{\pi}{\text{argmax}} \, \mathbb{E}_{s_0 \sim \mathcal{D}, \tau \sim (\pi, P)} \left[ \sum_{t=0}^{T-1} \mathbb{E}_{a \sim \pi(\cdot|s_t)}[r(s_t, a)] - \beta D_h(\pi(\cdot|\tau), \pi_{\text{ref}}(\cdot|\tau)) \right], \quad (7)$$

where $D_h$ is the Bregman divergence induced by a mirror map $h$. This new objective allows us to enforce different types of regularization, which, as we show later in the paper, can be tailored to account for specific properties of the preference dataset. Following the same intuition used to obtain the DPO objective, we have the following result.

**Theorem 3.1.** *Let $h_\phi$ be a $0$-potential mirror map and $\pi^\star$ be a solution to the optimization problem in (7). If $\pi_{\text{ref}}(a|s) > 0$ for all $s \in \mathcal{S}, a \in \mathcal{A}$, we have that*

$$r(\tau) = \phi^{-1}(\pi^\star(\tau)) - \phi^{-1}(\pi_{\text{ref}}(\tau)) + c(s_0), \quad (8)$$

*where $c(s_0)$ is a normalization constant that depends only on $s_0$.*

We provide a proof for Theorem 3.1 in Appendix B. By plugging (12) in the maximum likelihood problem in (3), we obtain the objective:

$$\pi^\star \in \underset{\pi}{\text{argmax}} \, \mathbb{E}_\mathcal{D} \left[ \log \sigma \left( \beta(\phi^{-1}(\pi(\tau_w)) - \phi^{-1}(\pi_{\text{ref}}(\tau_w)) - \phi^{-1}(\pi(\tau_l)) + \phi^{-1}(\pi_{\text{ref}}(\tau_l))) \right) \right], \quad (9)$$

---

[1] A function $h$ is *essentially smooth* if $\lim_{x \to \partial \mathcal{X}} \|\nabla h(x)\|_2 = +\infty$, where $\partial \mathcal{X}$ denotes the boundary of $\mathcal{X}$.

where $\mathbb{E}_{\mathcal{D}}$ is equivalent to $\mathbb{E}_{(s_0,\tau_w,\tau_l)\sim\mathcal{D}}$. When $\phi = e^x$, the expression in (7) recovers the DPO objective in (5). Additionally, we can modify the expression in (7) to include an SFT term in order to avoid the SFT step, in a manner similar to ORPO. Let $\psi$ be an $\omega$-potential and let $\pi_{\text{ref}}$ be the uniform distribution, then the new objective we consider is

$$\pi^\star \in \underset{\pi}{\arg\max}\, \mathbb{E}_{(s_0,\tau_w,\tau_l)\sim\mathcal{D}} \left[ \psi(\pi(\tau_w)) + \lambda \log \sigma \left( \phi^{-1}(\pi(\tau_w)) - \phi^{-1}(\pi(\tau_l)) \right) \right], \quad (10)$$

where the terms $-\phi^{-1}(\pi_{\text{ref}}(\tau_w))$ and $\phi^{-1}(\pi_{\text{ref}}(\tau_l))$ have canceled out due to $\pi_{\text{ref}}$ being uniform. We note that setting $\pi_{\text{ref}}$ to be the uniform distribution is equivalent to replacing the Bregman divergence penalty in (7) with the mirror map $h(\pi(\cdot|\tau))$, which enforces a form of entropy regularization. When $\psi(x) = \log(x)$ and $\phi^{-1}(x) = \log(x/(1-x))$, (10) recovers the ORPO objective in (6).

The objectives in (9) and (10) allow us to implement a wide variety of preference optimization algorithms, while benefiting from a theoretical justification. In the following, we will show that taking into account the properties of the preference dataset when choosing $\psi$ and $\phi^{-1}$ can lead to a better performance of the trained policy. Following the intuition developed by Jackson et al. (2024), we also allow our objective function, specifically $\psi$ and $\phi^{-1}$, to account for training progress. This is equivalent to changing the Bregman divergence penalty in (7) during training and, as we shall discuss in the following section, helps in dealing with datasets with mixed-quality data.

### 3.1 Learning mirror maps

To search the space of PO algorithms we have defined, we employ a neural network parametrization for both $\psi$ and $\phi^{-1}$, which we optimize using evolutionary strategies.

Similarly to Alfano et al. (2024), we parameterize both $\psi$ and $\phi^{-1}$ as a one layer neural network with 126 hidden units and non-negative kernels, where the activation functions are equally split among:

$$x,\ (x)^2_+,\ x^3,\ (x)^{1/2}_+,\ (x)^{1/3}_+,\ \log((x)_+),\ e^x,\ \tanh(x),\ \log(\text{clip}(x)/(1-\text{clip}(x))),$$

where $(x)_+ = \max(x,0)$ and $\text{clip}(x) = \max(\min(x,1),0)$. The non-negative kernels and the increasing activation functions guarantee the monotonicity of $\psi$ and $\phi^{-1}$, while the several different activation functions facilitate expressing complex functions. To ensure that we are able to recover the ORPO objective, we add $a\log(x)$ and $b\log(x/(1-x))$ to the final outputs of $\psi$ and $\phi^{-1}$, respectively, where $a, b \geq 0$. In case we want to take into account training progress, we give a second input to the neural network, i.e. $x \cdot n/N$, where $n$ is the current epoch and $N$ is the total number of epochs through the dataset. To ensure that monotonicity is preserved, we lower bound the weights associated to the second input with the negative of the respective weights associated to the first input.

To search for the best $\psi$ and $\phi^{-1}$ within this class, we employ the OpenAI-ES strategy (Salimans et al., 2017). Denote by $\zeta$ the parameters of $\psi$ and $\phi^{-1}$ and by $\pi^\zeta$ the final policy obtained optimizing the objective in (10) when using the parametrized $\psi$ and $\phi^{-1}$. Lastly, let $F(\zeta)$ be the expected cumulative reward of $\pi^\zeta$, i.e. $F(\zeta) = \mathbb{E}_{\tau\sim(\mu,\pi^\zeta,P)}r(\tau)$. We estimate the gradient $\nabla_\zeta F(\zeta)$ as

$$\mathbb{E}_{\epsilon\sim\mathcal{N}(0,I_d)} \left[ \frac{\epsilon}{2\sigma}(\widehat{F}(\zeta + \sigma\epsilon) - \widehat{F}(\zeta - \sigma\epsilon)) \right],$$

where $\mathcal{N}(0,I_d)$ is the multivariate normal distribution, $d$ is the number of parameters, $\widehat{F}$ is an estimate of $F$, and $\sigma > 0$ is a hyperparameter regulating the variance of the perturbations. We then use Adam (Kingma & Ba, 2015) to update the parameters $\zeta$ using the estimated gradient. In practice, to compute (13), we sample 128 values of $\epsilon$, obtain 256 perturbed objective functions. We then train 256 agents with the perturbed objective functions on an offline dataset. To obtain the value of each agent, i.e. $\widehat{F}(\zeta')$ for all perturbed $\zeta'$, we sample 100 trajectories on the target environment for each agent, and take the average cumulative reward as estimate for the value of the agent. Refer to Appendix D for further discussion and details on the ES methodology.

To accurately evaluate the agent's capabilities after training on the offline dataset, which may vary in quality, we generate and assess 100 trajectories in the original MuJoCo environment using the ground-truth MuJoCo reward function. This step ensures that the learned loss function generalizes effectively and mitigates known limitations of other PO functions, such as displacing probability mass away from the offline preference dataset (which would most likely result in poor performance when the agent is evaluated online). This evaluation provides a robust measure of the agent's generalization and highlights the adaptability of the optimized loss function.

Online evaluation is often resource-intensive, which has driven the popularity of offline algorithms like DPO. However, in this case, the cost of online evaluation is minimal due to the computational simplicity of the MuJoCo environment. Refer to Appendix F and Appendix E for further discussion on meta learning, online and offline algorithms.

## 4 EXPERIMENTS

We carry out our experiments on continuous RL tasks in MuJoCo and on LLM fine-tuning. To address resource constraints, we discover mirror maps in MuJoCo and report their performance when evaluated on a LLM finetuning task. To maximize computational efficiency, all our MuJoCo experiments are implemented in JAX (Bradbury et al., 2018) using the `brax` (Freeman et al., 2021), and evosax (Lange, 2022) libraries. For the LLM fine-tuning task, we modify the Alignment Handbook library (Tunstall et al.) to include our discovered objectives.

We consider three different preference optimization settings which we describe below. Each setting includes a starting policy, a set of preference datasets with different properties and an evaluation method for the trained policy. We apply ES to the first two settings (MuJoCo) and test the discovered objectives on the last one (LLM fine-tuning).

### 4.1 DATASET TYPES

Across all analysed MuJoCo environments, we generate several datasets to test the efficacy and robustness of our discovered loss functions. Our datasets are meant to model common issues of real world data. The dataset types are described below:

- **Base dataset**: we compare a trajectory from the expert agent with one of a reference agent with varying skill levels.
- **Noisy dataset**: we flip a varying portion of the rankings in the base dataset.
- **Shuffled dataset**: the comparisons are not always between the expert agent and a second agent. Namely, 25% of the comparisons are between two trajectories from the expert agent, 50% of the comparisons are between a trajectory of the expert agent and one of a reference agent, and 25% of the comparisons are between two trajectories of a reference agent.
- **Poor judge dataset**: the judge is more likely to flip labels when the two trajectories have closer reward values. This is implemented as an increase in the temperature of the Bradley-Terry judge.

### 4.2 ENVIRONMENTS

In MuJoCo, we analyze two environments: `Hopper` and `Ant`. Our objective in `Hopper` is to learn a policy from scratch, i.e. with a randomly initialized policy, using a preference dataset. However, this setting is not representative of the typical conditions of LLM fine-tuning tasks, which usually involve a pretrained model. To address this gap, we consider a setting where a pretrained agent is provided, and the task is to adapt its behavior to meet the original objective while adhering to an additional stylistic constraint. Specifically, in the `Ant` environment, we start from an agent that has been pre-trained on the standard `Ant` goal of moving forward, and enforce the objective of avoiding the use of one of its legs. This is accomplished by introducing the `Three-legged-ant` (TLA) environment, a modified version of `Ant` where utilizing the fourth leg results in significant penalties. The trained policy is then evaluated on the TLA environment.

**Hopper**  On `Hopper`, we train four reference agents of different skill levels, with respective expected cumulative reward of 900, 1200, 1800, and 2100 (the expert agent). Each dataset consists of 5120 rows, each with two trajectories of length 1000 starting from the same state. We generate a dataset by comparing trajectories by the expert agent, with other trajectories by one of our reference agents. A Bradley-Terry judge ranks each pair of trajectories, based on their true reward.

**Ant**  We train one agent in the original `Ant` environment, achieving a reward of 6000, and another in the `TLA` environment, which achieves a reward of 3900. For comparison, the agent trained in

Table 1: Results for `Hopper`. For each dataset, we report the average value and standard error of 25 agents trained using ORPO and the objective discovered for that dataset. When ranking two trajectories of value 2100 and 1800, a judge with low, medium, and high temperature provides the correct ranking 95%, 85% and 75% of the time, respectively.

| Value agent 1 | Value agent 2 | Noise | Shuffled | Judge temp. | ORPO | Discovered |
|---|---|---|---|---|---|---|
| 2100 | 900 | 0 | No | Low | 2003±20 | 2012±11 |
| 2100 | 1200 | 0 | No | Low | 2055±12 | 2055±10 |
| 2100 | 1800 | 0 | No | Low | 2043± 15 | 2073±12 |
| 2100 | 2100 | 0 | No | Low | 2070±12 | 2098±11 |
| 2100 | 900 | 0.1 | No | Low | 1519±45 | **1724±43** |
| 2100 | 1800 | 0.1 | No | Low | 1936±25 | 1945±25 |
| 2100 | 2100 | 0.1 | No | Low | 2112±9 | 2094±13 |
| 2100 | 900 | 0.2 | No | Low | 662±49 | **1127±45** |
| 2100 | 900 | 0.3 | No | Low | 623±29 | **698±26** |
| 2100 | 1800 | 0 | No | Medium | 1884±22 | 1928±23 |
| 2100 | 1800 | 0 | No | High | 1858±32 | 1895±14 |
| 2100 | 900 | 0 | Yes | Low | 975±51 | **1099±35** |

Table 2: Results for `TLA`. For each dataset, we report the average value and standard error of 25 agents trained using ORPO and the objective discovered for that dataset.

| Value agent 1 | Value agent 2 | Noise | Shuffled | ORPO | Discovered |
|---|---|---|---|---|---|
| 3900 | 1700 | 0 | No | 3277±49 | **3473±63** |
| 3900 | 1700 | 0 | Yes | 2425±58 | **3485±163** |
| 3900 | 1700 | 0.1 | No | 2675±62 | **3837±52** |

the original `Ant` environment achieves a reward of 1700 when evaluated in the `TLA` environment. The dataset generation follows the same protocol as described for the `Hopper` environment, where trajectories are collected from both the `Ant` and `TLA` agents. However, in this setting the number of rows for each dataset is 1280, to account for the fact that we do not want the agent to learn from scratch, but to adjust its policy to a new instruction.

**LLM tuning** Finally, we evaluate one of our discovered objective functions on a real-world LLM fine-tuning task. To simulate a scenario involving mixed data quality, we fine-tune the gemma-7b (Team et al., 2024) model on a modified version of the dpo-mix-7k[2] dataset, where half of the responses, selected at random, are replaced with responses generated by gemma-2b (Team et al., 2024), a model that typically produces lower-quality responses compared to those originally present in DPO-mix. This approach replicates the shuffled dataset described for the MuJoCo experiments, aiming to simulate the challenges of training with datasets of varying quality, a common issue in fine-tuning tasks where it is difficult and costly to ensure uniformly high-quality data.

We use the same hyper-parameters for all MuJoCo datasets, which have been tuned for ORPO using Weights and Biases (Biewald, 2020). We report them in Appendix G, along the hyper-parameters for OpenES. For the Alignment Handbook library, we use the default hyperparameters.

## 5 RESULTS

We provide the results of our experiments for MuJoCo in Tables 1, 2, and 3, which report the performance of ORPO and our discovered objectives for `Hopper`, `TLA`, and the temporally aware case, respectively. To provide information on the behavior of these objectives as well as guidelines on how to design loss landscapes for the settings we have considered, we compare the landscapes of ORPO and of the discovered objectives in Figures 1, 2, 3, and 4. In these figures, we report, in absolute value, the gradients of the objective with respect to $\log(\pi(\tau_w))$ and $\log(\pi(\tau_l))$, as well as a sampled training trajectory. We discuss the results below, by dataset type.

---

[2]https://huggingface.co/datasets/argilla/dpo-mix-7k

Table 3: Results for temporally-aware objectives. For each dataset, we report the average value and standard error of 25 agents trained using ORPO and the temporally-aware objective discovered for that dataset.

| Value agent 1 | Value agent 2 | Env | Noise | Shuffled | ORPO | Discovered |
|---|---|---|---|---|---|---|
| 2100 | 900 | Hopper | 0 | Yes | 975±51 | **2032±13** |
| 3900 | 1700 | TLA | 0 | No | 3277±49 | **3903±42** |
| 3900 | 1700 | TLA | 0 | Yes | 2425±58 | **3251±83** |

While we benchmarked our approach against DPO and Conservative DPO (Mitchell, 2023), we do not report the results as both methods failed in all tested scenarios. This failure is consistent with observations in prior work (Xu et al., 2024b), which demonstrated that DPO can move probability mass away from the available datapoints, leading to catastrophic performance. This behavior appears to be particularly pronounced in the MuJoCo setting due to our agent's architecture, whereby the policy is parametrized as a Gaussian distribution and moving the mean is enough to cause displacement. These results underscore the robustness of our method in identifying failure cases and limitations of existing PO loss functions. In practical applications, such as fine-tuning LLMs, complex interactions among various components often obscure these behaviors. Notably, the issues inherent to DPO became evident only after extensive analysis, as they were not immediately apparent in LLM tuning experiments.

## 5.1 BASE DATASET

For the base dataset, we observe that the quality of the final agent is only marginally affected by the quality of the rejected datapoints and that the final performance never exceeds the performance of the expert agent. For the Hopper dataset, we see that ORPO achieves the performance of the best reference agent in all cases and that it is not significantly outperformed by the learned objective. However, ORPO does not reach the performance of the best reference agent in the TLA dataset and is outperformed by our discovered objectives. Remarkably, as shown in Table 3, the temporally aware objective achieves the performance of the expert agent, with a 19% improvement over ORPO.

As shown by Figures 1 and 4, both the static and the temporally aware discovered objectives present a smaller gradient than ORPO, which means that they cause smaller updates. Additionally, for ORPO we have that if $\log(\pi(\tau_w)) < \log(\pi(\tau_l))$ then the gradients increases suddenly, while for the temporally aware discovered objective the gradient increases more slowly.

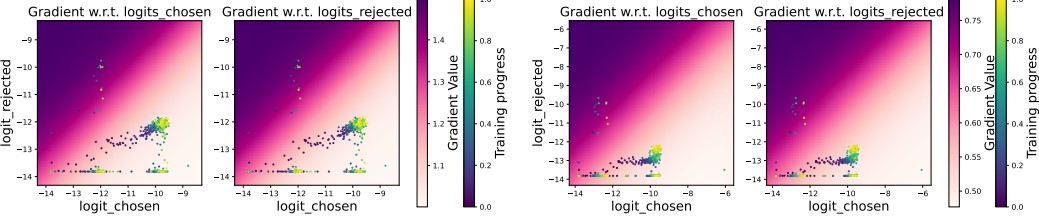

Figure 1: **Base Hopper (left) and TLA (right) datasets**. The level of noise is 0.1 for both plots. Values are absolute gradient values and the dots are a randomly sampled subset of training data.

## 5.2 NOISY DATASET

As expected, the performance of ORPO declines when the dataset contains randomly flipped labels, with the degradation becoming more pronounced as the percentage of flipped labels increases. While the performance degrades for our discovered objectives in Hopper as well, our discovered loss functions show significant improvements over ORPO across all noise levels. In TLA, the discovered objective is capable of almost matching the expert agent performance despite the noise.

Figure 2 shows that the objective discovered in `Hopper` assigns a roughly constant value to the gradient w.r.t. the chosen logits, while the gradient w.r.t. the rejected logits increases as $\log(\pi(\tau_w)) - \log(\pi(\tau_l))$ decreases.

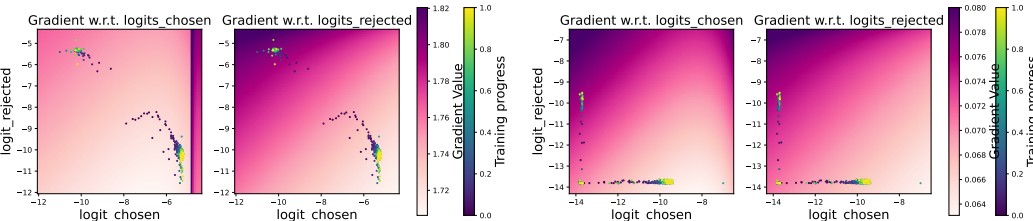

Figure 2: **Noisy `Hopper` (left) and `TLA` (right) datasets**. The level of noise is 0.1 for both plots. Values are absolute gradient values and the dots are a randomly sampled subset of training data.

## 5.3 SHUFFLED DATASET

While using the same data as the base dataset, the performance of ORPO significantly decreases in both the `Hopper` and TLA shuffled dataset, where low quality trajectories are selected as the preferred choice when compared to other low quality trajectories. In contrast, the discovered objectives achieve a value of the final policy close to the best reference agent. In particular, the value reached by the temporally aware objective for `Hopper` is more than double the value reached by ORPO. Overall, all discovered objectives present smaller gradients than ORPO, and therefore induce smaller updates, which prevents taking large updates when the chosen logits belong to a low-quality trajectory.

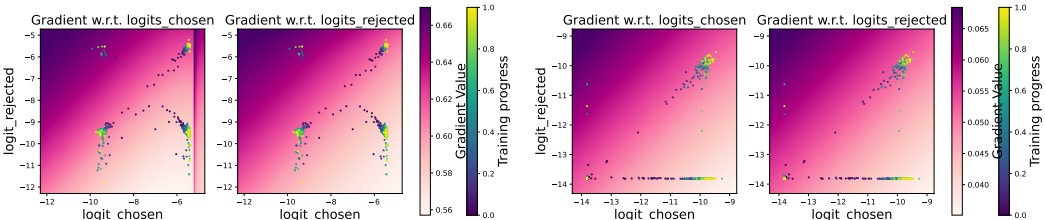

Figure 3: **Shuffled `Hopper` (left) and `TLA` (right) datasets**. Values shown are absolute gradient values and the dots are a randomly sampled subset of training data.

## 5.4 POOR JUDGE

As expected, we observe a decrease in performance for ORPO when we increase the temperature of the judge. However, since we compared trajectories from two high-quality reference agents, the drop in performance is small. This is also reflected in the discovered objectives, which do not manage to significantly outperform ORPO.

## 5.5 TRANSFER

We show that PO objectives learnt with our methodology can transfer to other domains. In particular, the static objective discovered on the shuffled TLA dataset tranfers to both the shuffled `Hopper` dataset, where it obtains a final policy value of 1735±39, and to the LLM-tuning task with shuffled data defined above. We train the base LLM on the modified DPO-mix dataset with both ORPO and the discovered objective. We observe that ORPO achieves 57% accuracy on the test set while our discovered algorithm achieves 62%. Finally, we compare the two trained models with AlpacaEval (Li et al., 2023), and obtain a 53% winrate for the model trained with the discovered objective.

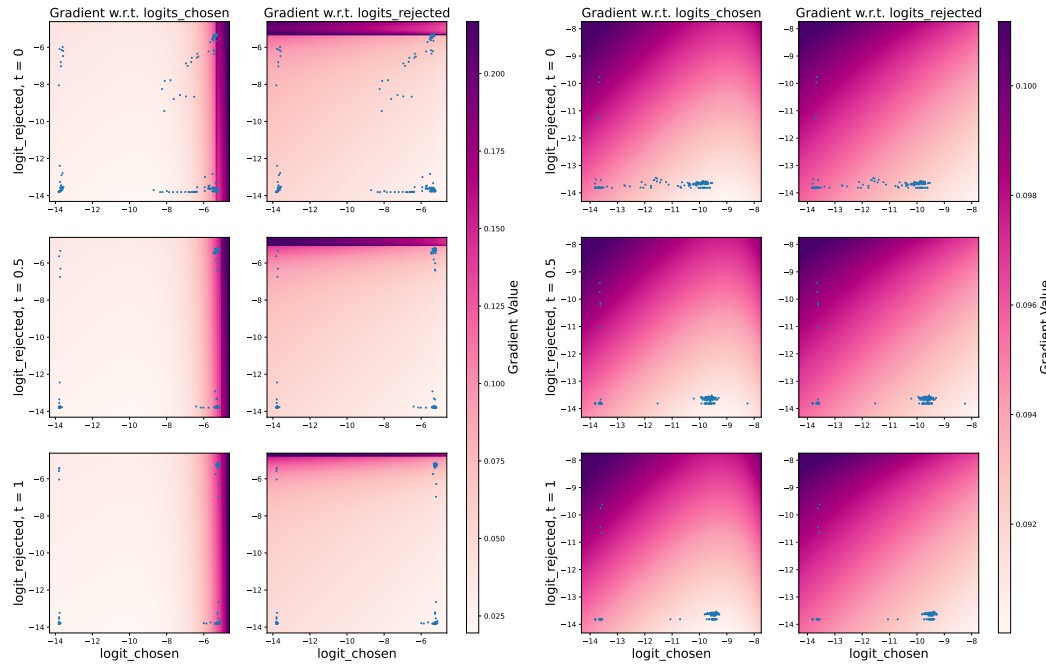

Figure 4: **Shuffled `Hopper` (left) and `TLA` (right) datasets - temporally aware objective**. Gradients are shown at three different moments of the training progress. For each row, the dots represent the respective third of a sampled training trajectory.

### 5.6 FORMALIZING THE DISCOVERED OBJECTIVE

Via regression on a few basis functions, we find a simple expression that closely approximates the objectives discovered on the shuffled and noisy `TLA` datasets and leads to similar performance. Letting $\psi(x) = \alpha \log(x)$ and $\phi^{-1}(x) = \beta \log(x)$, we have that the objective in (10) becomes

$$\pi^{\star} \in \operatorname*{argmax}_{\pi} \mathbb{E}_{(s_0, \tau_w, \tau_l) \sim \mathcal{D}} \left[ \alpha \log(\pi(\tau_w)) + \lambda \log \sigma \left( \beta(\log(\pi(\tau_w) - \log(\pi(\tau_l))) \right) \right]. \quad (11)$$

For specific values of $\alpha$ and $\beta$, this closed-form expression effectively captures the behavior and performance of the original objectives. This simplification reduces computational overhead and enhances interpretability, making the objective easier to implement in practical settings.

## 6 CONCLUSION

We have introduced a novel framework for Policy Optimization algorithms, as well as a methodology for the automatic discovery of PO algorithms using evolution strategies. Through a systematic evaluation across diverse settings in MuJoCo environments, we demonstrated that our discovered objective functions consistently match or exceed the performance of existing methods, particularly in noisy and mixed-quality datasets where the ORPO baseline struggles. The introduction of temporally aware objective functions further improved performance, allowing the optimization process to vary and adapt during training. Analyzing the landscape of the discovered objectives, we give an intuition that justifies improved performance as well as guidance in the design of new PO algorithms. Our results also indicate that the discovered objectives generalize beyond simple reinforcement learning environments, showing promising performance when transferred to LLMs, thereby confirming the broader applicability of our approach.

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

## A  RELATED WORK

**Automatic Discovery of Preference Optimization Loss Functions**   Several works in the litera-
ture have shown that it is possible to discover machine learning algorithms that outperform algo-
rithms manually designed by researchers (Oh et al., 2020; Lu et al., 2022; Jackson et al., 2024;
Alfano et al., 2024). An approach particularly relevant to our method is DiscoPOP by Lu et al.
(2024), which leverages an LLM to discover objective functions for LLM tuning. They consider a
different space of objective functions from us, as they replace the log-sigmoid in (5) with a generic
loss function, following the framework built by Tang et al. (2024). Additionally, instead of searching
over a space of parametrized functions, they ask the LLM to generate loss functions in code space.
This distinction suggests that our approaches could be complementary, as the model discovered by
DiscoPOP could be paired with our learned mirror map. Lastly, DiscoPOP optimizes its objective
function directly on the final task, whereas we adopt a two-stage process—optimizing the loss func-
tion on a separate task (MuJoCo) and later transferring it to the LLM setting. This transferability
underscores the broader applicability of our approach.

**Generalisations of DPO**   A generalization of DPO alternative to ours is $f$-DPO, developed
by Wang et al. (2023), which consists in replacing the KL-divergence in (2) with an $f$-divergence
and then apply the same heuristic as DPO to obtain the final objective function. We note that the
KL-divergence is the only $f$-divergence to be also a Bregman divergence, and vice-versa. They
empirically demonstrate that different $f$-divergences lead to different balances between alignment
performance and generation diversity, highlighting the trade-offs inherent to this class of algorithms.
Huang et al. (2024) further explore this class of PO algorithm and individuate an $f$-divergence for
which $f$-DPO is robust to overoptimization.

## B  PROOF OF THEOREM 3.1

We provide here a proof for our main result, i.e. Theorem 3.1. The proof to obtain the DPO objective
in (5) follows by taking $\phi = e^x$.

**Theorem B.1** (Theorem 3.1)**.** *Let $h_\phi$ be a 0-potential mirror map and $\pi^\star$ be a solution to the
optimization problem in* (7)*. If $\pi_{\mathrm{ref}}(a|s) > 0$ for all $s \in \mathcal{S}, a \in \mathcal{A}$, we have that*

$$r(\tau) = \phi^{-1}(\pi^\star(\tau)) - \phi^{-1}(\pi_{\mathrm{ref}}(\tau)) + c(s_0), \tag{12}$$

*for all trajectories $\tau$, where $c(s_0)$ is a normalization constant that depends only on $s_0$.*

*Proof.* We use the KKT conditions to solve (7), i.e.

$$\pi^\star \in \underset{\pi}{\arg\max} \, \mathbb{E}_{s_0 \sim \mathcal{D}, \tau \sim (\pi, P)} \left[ \sum_{t=0}^{T-1} \mathbb{E}_{a \sim \pi(\cdot|s_t)}[r(s_t, a)] - \beta D_h(\pi(\cdot|\tau), \pi_{\mathrm{ref}}(\cdot|\tau)) \right]$$

We use the stationarity condition to obtain the equation

$$\nabla_{\pi(\tau)} \left[ \sum_{t=0}^{T-1} \mathbb{E}_{a \sim \pi(\cdot|s_t)}[r(s_t, a)] - \beta D_h(\pi(\cdot|\tau), \pi_{\mathrm{ref}}(\cdot|\tau)) \right.$$

$$\left. - \lambda \left( \sum_{\tau':s_0 \in \tau'} \pi(\tau') - 1 \right) + \sum_{\tau':s_0 \in \tau'} \alpha(\tau')\pi(\tau') \right]$$

$$= r(\tau) - \beta\phi^{-1}(\pi(\tau)) + \phi^{-1}(\pi_{\mathrm{ref}}(\tau)) - \lambda + \alpha(\tau) = 0,$$

for all initial states $s_0 \in \mathcal{S}$ and for all trajectories $\tau$ starting from $s_0$. Rearranging, we obtain that

$$\pi(\tau) = \phi\big(r(\tau) + \phi^{-1}(\pi_{\mathrm{ref}}(\tau)) - \lambda + \alpha(\tau)\big).$$

Since $0 \notin \mathrm{dom}\,\phi^{-1}$, due to the definition of a 0-potential, and $\pi_{\mathrm{ref}}(\tau) > 0$, we have that $\pi(\tau) > 0$
for all trajectories $\tau$. Invoking the complementary slackness condition, whereby $\alpha(\tau)\pi(\tau) = 0$ for
all trajectories $\tau$, we have that $\alpha(\tau) = 0$ for all trajectories $\tau$. Therefore, we have that

$$r(\tau) - \beta\phi^{-1}(\pi(\tau)) + \phi^{-1}(\pi_{\mathrm{ref}}(\tau)) - \lambda = 0$$

The theorem statement is obtained by rearranging the last equation and denoting $c(s_0) = \lambda$    $\square$

## C    FURTHER DISCUSSION OF $\omega$-POTENTIALS

We show here two examples of Bregman divergence induced by an $\omega$-potential mirror map, that is when $\phi(x) = e^{x-1}$ and when $\phi(x) = x$. If $\phi(x) = e^{x-1}$, the associated mirror map is defined as

$$
\begin{aligned}
h_\phi(\pi(\cdot|s)) &= \sum_{a \in \mathcal{A}} \int_1^{\pi(a|s)} \phi^{-1}(x) dx = \sum_{a \in \mathcal{A}} \int_1^{\pi(a|s)} (\log(x) + 1) dx \\
&= \sum_{a \in \mathcal{A}} \pi(a \mid s) \log(\pi(a \mid s)) - \pi(a \mid s) + \pi(a \mid s) \\
&= \sum_{a \in \mathcal{A}} \pi(a \mid s) \log(\pi(a \mid s)),
\end{aligned}
$$

which is the negative entropy. Plugging this expression in the definition of Bregman divergence we obtain

$$
\begin{aligned}
\mathcal{D}_h(x, y) &= h(x) - h(y) - \langle \nabla h(y), x - y \rangle \\
&= \sum_{a \in \mathcal{A}} x_a \log(x_a) - y_a \log(y_a) - (\log(y_a) - y_a)(x_a - y_a) \\
&= \sum_{a \in \mathcal{A}} x_a \log(x_a/y_a),
\end{aligned}
$$

which is the definition of the KL-divergence. If $\phi(x) = 2x$, the associated mirror map is defined as

$$
\begin{aligned}
h_\phi(\pi(\cdot|s)) &= \sum_{a \in \mathcal{A}} \int_1^{\pi(a|s)} \phi^{-1}(x) dx = \sum_{a \in \mathcal{A}} \int_1^{\pi(a|s)} 2x dx \\
&= \sum_{a \in \mathcal{A}} \pi(a \mid s)^2,
\end{aligned}
$$

which is the $\ell_2$-norm. Plugging this expression in the definition of Bregman divergence we obtain

$$
\begin{aligned}
\mathcal{D}_h(x, y) &= h(x) - h(y) - \langle \nabla h(y), x - y \rangle \\
&= \sum_{a \in \mathcal{A}} x_a^2 - y_a^2 - (2y_a)(x_a - y_a) \\
&= \sum_{a \in \mathcal{A}} (x_a - y_a)^2,
\end{aligned}
$$

which is the definition of the Euclidean distance.

## D    FURTHER DISCUSSION ON EVOLUTION STRATEGIES

Evolution Strategies (ES) represent a powerful, backpropagation-free method for optimizing complex functions, that has been particularly successful in the context of long-horizon, noisy, and bi-level optimization tasks such as RL and meta-RL. ES, and in particular the OpenAI-ES algorithm (Salimans et al., 2017), rely on perturbation-based sampling to estimate gradients without requiring backpropagation through the entire computational graph. This feature makes ES well-suited for tasks with long computational graphs, for instance algorithms with many updates, where, due to memory constraints, traditional gradient-based methods have to resort to gradient truncation, introducing bias (Werbos, 1990; Metz et al., 2022; Liu et al., 2022).

In our setting, we use ES to search for the best $\psi$ and $\phi^{-1}$ within the parametrized class introduced in Section 3.1, so that an agent trained using the objective in (11) achieves the highest value. Denote by $\zeta$ the parameters of $\psi$ and $\phi^{-1}$ and by $\pi^\zeta$ the final policy obtained optimizing the objective in (10) when using the parametrized $\psi$ and $\phi^{-1}$. Lastly, let $F(\zeta)$ be the expected cumulative reward (or value) of $\pi^\zeta$, i.e. $F(\zeta) = \mathbb{E}_{\tau \sim (\mu, \pi^\zeta, P)} r(\tau)$. At each iteration, we estimate the gradient $\nabla_\zeta F(\zeta)$ as

$$
\mathbb{E}_{\epsilon \sim \mathcal{N}(0, I_d)} \left[ \frac{\epsilon}{2\sigma} (\widehat{F}(\zeta + \sigma\epsilon) - \widehat{F}(\zeta - \sigma\epsilon)) \right], \tag{13}
$$

where $\mathcal{N}(0, I_d)$ is the multivariate normal distribution, $d$ is the number of parameters, $\widehat{F}$ is an estimate of $F$, and $\sigma > 0$ is a hyperparameter regulating the variance of the perturbations.

## E   FURTHER DISCUSSION ON ONLINE VS OFFLINE METHODS

In the domain of RL and preference optimization, the choice between online and offline algorithms presents a critical trade-off, influencing computational efficiency, data requirements, and generalization capabilities. Online methods, such as PPO, iteratively collect and incorporate new data during training. These inherently support exploration of the environment, enabling the discovery of novel strategies or behaviors that are not captured in pre-existing datasets. However, they need feedback for each generated "trajectory" (or response, in the LLM case), which might be expensive to obtain. Online methods are also more complex and particularly sensitive to hyperparameters, often requiring meticulous tuning for stability and efficiency.

Offline algorithms, such as DPO and its variants, rely entirely on pre-collected datasets. These methods are designed for efficiency and simplicity: they don't require any additional feedback from users and are therefore particularly effective in scenarios where feedback is delayed or unavailable. However, the reliance on static datasets means offline methods may struggle to generalize beyond the training data, particularly if the distribution shift between the training dataset and test time distribution is significant. Additionally, the performance of the algorithm is closely tied to the quality of the training dataset: noisy, biased, or corrupt datasets can severely degrade performance, as these methods cannot mitigate such issues through exploration or resampling.

In summary, RLHF (i.e., online) is considered the superior approach, particularly when substantial amounts of online labels are accessible. This makes it the industry standard (Xu et al., 2024b). While DPO has been theoretically equated to optimizing using PPO and a reward model trained on an offline dataset, recent empirical research (Tang et al., 2024) has challenged this notion. These studies have demonstrated that online methods, such as PPO, consistently outperform offline methods like DPO. This superiority is attributed to the benefits of on-policy sampling.

While DPO has occasionally outperformed PPO, it's important to note that several studies (Xu et al., 2024b; Song et al., 2024) have consistently shown PPO's overall superiority. DPO's relative strength lies in its simpler training regime, which avoids the complexities associated with reward model inaccuracies. However, DPO's performance is significantly limited by its sensitivity to distribution shift, especially when the offline preference data lacks diversity (Song et al., 2024). This limitation becomes particularly evident when querying the model with out-of-distribution data, a common challenge for methods relying solely on offline data. To mitigate this issue, DPO-iter (Xu et al., 2024b), which incorporates online data, has been proposed as a potential solution.

## F   FURTHER DISCUSSION ON META-LEARNING ALGORITHMS

Meta-learning, or "learning to learn", has been extensively employed to automate the design of algorithms that can either adapt rapidly with minimal data samples or generalize effectively to unseen data, tasks, or environments. The development of broadly applicable algorithms is particularly critical in the context of preference optimization for LLMs. Here, LLMs are fine-tuned on relatively small datasets of offline data but must generalize to a virtually infinite range of potential user queries. Prior work in meta-learning has demonstrated success in developing generalizable optimization algorithms and loss functions (Lu et al., 2022; Jackson et al., 2024; Lu et al., 2024; Goldie et al., 2024; Kirsch et al., 2020).

At its core, meta-learning is defined as a bilevel optimization problem with an inner and an outer loop. The inner loop consists in an iterative optimization algorithm that trains agents to solve a predetermined task given a set of meta-parameters. The outer loop consists in evaluating the agents trained in the inner loop and update the meta-parameters accordingly, following some optimization method like second order gradient descent (Finn et al., 2017). The evaluation of the agents is typically done on a held-out dataset in supervised learning or by sampling trajectories on the environment simulator in RL (Lu et al., 2022; Jackson et al., 2024). In our setting, the inner loop is the offline preference optimization algorithm, while the outer loop is the agent evaluation on the environment (online) and the update of the meta-parameters $\zeta$.

# G HYPER-PARAMETERS

We give the hyper-parameters we use for training in Tables 4 and 5.

| Parameter | Value |
| --- | --- |
| Number of epochs | 12 |
| Minibatch size | 2 |
| Learning rate | 1e-3 |
| Max gradient norm | 1.3 |
| $\lambda$ | 0.5 |

Table 4: Hyper-parameter settings for PO.

| Parameter | Hopper | TLA |
| --- | --- | --- |
| Population Size | 256 | 256 |
| Number of generations | 128 | 256 |
| Sigma init | 0.03 | 0.03 |
| Sigma Decay | 0.999 | 0.999 |
| Learning rate | 0.02 | 0.02 |

Table 5: Hyper-parameter settings of OpenAI-ES.

