# OpenReview forum: "Learning Loss Landscapes in Preference Optimization"
_ICLR.cc/2025/Conference — Submitted to ICLR 2025_

### Official Review · Reviewer_cDRz · 2024-10-28

**Soundness:** 1
**Presentation:** 2
**Contribution:** 1
**Rating:** 3
**Confidence:** 4

**Summary:**

This paper generalizes a recent variant of DPO, namely ORPO, by introducing two maps into the problem definition of ORPO. Then these maps are parametrized by a neural network that is trained by OpenAI-ES approach. An empirical study shows better performance than the original ORPO in some settings of noisy preference datasets and competitive performance in other settings.

**Strengths:**

It generalizes a recent and promising variant of DPO, namely ORPO, by introducing mirror maps.

Empirical evaluation shows that optimizing the parameterized mirror maps by OpenAI-ES sometimes results in better performance.

**Weaknesses:**

The position of this study among related studies has not been clarified. Several studies have investigated the preference optimization under noisy preferences and proposed variants of DPO, such as cDPO and rDPO. I see that the current paper is based on ORPO and hence the direct comparison may not be fully fair, but at the same time, I do not see the advantages of the current approach. On the other hand, there are several studies modifying the loss or the divergence in DPO such as IPO and f-DPO. The novelty of this study is not clearly stated among these related works. The literature review should cover these topics, however, in this paper, only f-DPO is mentioned in the related work as a generalization of DPO.

The motivation is not stated clearly. Why is the generalization of ORPO as in (10) and searching the introduced maps expected to improve the performance of ORPO for noisy preference datasets? I could not see the logic behind it. Because the logic is not provided, I do not see whether the experimental results are really associated with the generalization. Does this generalization contribute to the good performance? From the current experiments, I do not see whether the proposed approach is better than simply introducing some scalar factors to all terms in ORPO and optimizing them.

Reproducibility is not high. First, the proposed learning scheme for mirror maps is not clearly stated. It requires estimating the cumulative reward for each obtained policy, but how is it estimated in an offline manner (stated in Line 102)? Second, the code to reproduce the experiments is not available. Some details, such as the architecture of the policy network and temperature parameters.

Comparison to other approaches is missing. The proposed approach is compared only to the original ORPO, which is not designed to address noisy preference datasets. No comparisons with DPO variants such as rDPO specialized for noisy preference datasets are provided.

**Questions:**

Please address the questions given in the weaknesses section.

Additional comments are listed here:

Additional comments are listed here:

In the experiments in Table 1, what is the difference between setting a high temperature and adding a noise in this setting? They look like the same.

How is the temperature parameter in Table 2 set?

How are the obtained mirror maps interpreted? If it is transferable, shouldn’t the result be analyzed?

In line 222, it was not clear to me how these terms canceled out. Could you elaborate on the derivation?

---

> ### Author Response · Authors · 2024-11-21
>
> We thank the review for the time spent reviewing our paper and their valuable feedback. Addressing the concerns raised:
>
> > The position of this study among related studies has not been clarified. Several studies have investigated the preference optimization under noisy preferences and proposed variants of DPO, such as cDPO and rDPO.
>
> We have updated the abstract and introduction to make the positioning of our paper clearer, but summarising, our paper wants to provide a framework for practitioners to be able to test, gain intuition and discover loss functions customised to their specific setting. For many researchers, including us, it is computationally infeasible to perform thorough analysis and comparisons in an LLM setting. Additionally, results might be confusing or difficult to interpret, due to complex interactions across components and the challenges in evaluating the final performance. Therefore, we propose a framework to perform analysis on simpler RL environments, namely MuJoCo, where rewards and final agent ability are objective and easy to evaluate, and show our discovered loss transfers to the LLM setting.
> We have benchmarked agains cDPO (and added this discussion to the paper) but all DPO variants catastrophically failed in our setting, due to the tendency of DPO to move probability mass away from the given dataset. Moreover, we believe comparison to custom-losses hand-crafted for noisy datasets is not appropriate, as our framework is trying to present a way to automatically discover a custom loss function for any dataset characteristic, not just noisy datasets. In the paper we report several such datasets, but researchers could analyse others that are particularly relevant to their specific problem setting.
>
> Regarding the rDPO comparison, although we agree data augmentation and other similar methods would absolutely be beneficial in a noisy setting, we believe this is beyond the scope of our work, as such methods are orthogonal to our proposed framework.
>
> > Why is the generalization of ORPO as in (10) and searching the introduced maps expected to improve the performance of ORPO for noisy preference datasets? I could not see the logic behind it. Because the logic is not provided, I do not see whether the experimental results are really associated with the generalization. Does this generalization contribute to the good performance? From the current experiments, I do not see whether the proposed approach is better than simply introducing some scalar factors to all terms in ORPO and optimizing them.
>
> - **Assumptions behind the theoretical derivation**: We note that the assumptions behind Theorem 3.1 are: (1) the reference policy needs to be positive for each state action pair, i.e. $\pi_\text{ref}(a|s)>0$ for all $s\in\mathcal{S}, a\in\mathcal{A}$, and (2) the mirror map $h_\phi$ needs to be a 0-potential. The first assumption is always respected in LLM tuning tasks, where the probability distribution induced by the LLM is parametrized by a softmax head that assigns positive probability to all tokens. The second assumption can also be easily satisfied, as we can enforce any mirror map to be a 0-potential mirror map by adding the negative entropy mirror map. That is, for any mirror map $h$ and any constant $\alpha>0$, we have that $h'(x)= h(x)+\alpha\sum_a \log(x_a)$ is a 0-potential mirror map.
>  - **Motivation behind searching the objective family in (10)**: Obtaining generalized families of established algorithms and searching for alternative members of the family is a popular research direction in statistics and machine learning. In RL, examples are Policy Mirror Descent [1] and Mirror Learning [2,3]. In RLHF, an example is $DPO-f$ [4]. In optimization, Mirror Descent starts as a generalization of gradient descent but several of its versions have been shown to improve over gradient descent [5,6]. In general, different members in the generalized family present different characteristics, induce a different optimization landscapes, and are less/more suited for specific scenarios. By defining a wide class of algorithms, as we do in equation (10), we increase the chance of being able to find a member of the class that is capable of outperforming ORPO. On the other hand, if we simply apply scalar coefficients within the ORPO objective, we restrict the class of algorithm we are able to search and reduce the chance of finding new more performing algorithms.

---

> ### Author Response · Authors · 2024-11-21
>
> > Reproducibility is not high. First, the proposed learning scheme for mirror maps is not clearly stated. It requires estimating the cumulative reward for each obtained policy, but how is it estimated in an offline manner (stated in Line 102)? Second, the code to reproduce the experiments is not available. Some details, such as the architecture of the policy network and temperature parameters.
>
> Here is a link to our anonymised code to ensure reproducibility: https://anonymous.4open.science/r/MuJoCo_preferences-B7A4/
> We also commit to adding a public link to the paper for the camera ready version.
> The cumulative reward is estimated online. We have updated the referenced line in the preliminaries section to make it more clear:
> "In this work we consider an offline setting during training, where we do not have access to either the transition probability $P$, the reward function $r$, or the MDP $M$. During evaluation, we calculate cumulative rewards online."
>
> > Comparison to other approaches is missing. The proposed approach is compared only to the original ORPO, which is not designed to address noisy preference datasets. No comparisons with DPO variants such as rDPO specialized for noisy preference datasets are provided.
>
> We have updated the paper to discuss our comparison to DPO baselines.
>
> Responding to the questions:
>
> > In the experiments in Table 1, what is the difference between setting a high temperature and adding a noise in this setting? They look like the same.
>
> These are very similar settings but with a slight difference, when the judge temperature is high there is a higher likelihood of assigning an incorrect label if the rewards are close to each other (i.e. if the two datapoints are similar). With the noisy dataset, the probability of assigning an incorrect label is independent of how similar or different the rewards of the two datapoints are.
>
> > How is the temperature parameter in Table 2 set?
>
> In the experiments in table 2, there is no judge temperature (= 0.0).
>
> > How are the obtained mirror maps interpreted? If it is transferable, shouldn’t the result be analyzed?
>
> Yes, thank you for your helpful request, we added more analysis in the paper and we now have a closed formed expression for our discovered loss function. Summarising our analysis, the loss functions benefit from a smoother and smaller gradient in the noisy and shuffled setting (where low quality datapoints are sometimes selected as the “preferred” choice, even though the preferred/rejected label is correct, as the datapoint is compared to worst performing one). Particularly, in the shuffled dataset, we also notice a low coefficient on the SFT term compared to the PO term helps performance. This is intuitive as the quality of the preferred choices is not consistent, so the SFT term should be downweighted.
>
> > In line 222, it was not clear to me how these terms canceled out. Could you elaborate on the derivation?
>
> We have updated the paper to include a more detailed explanation. Reporting it here:
>
> "The terms $-\phi^{-1}(\pi_{\mathrm{ref}}(\tau_w))$ and $\phi^{-1}(\pi_{\mathrm{ref}}(\tau_l ))$ have canceled out due to $\pi_{\mathrm{ref}}$ being uniform. We note that setting $\pi_{\mathrm{ref}}$ to be the uniform distribution is equivalent to replacing the Bregman divergence penalty in (7) with the mirror map $h(\pi(\cdot|\tau))$, which enforces a form of entropy regularization. When $\psi(x) = \log(x)$ and $\phi^{-1}(x) = \log(x/(1-x))$, (10) recovers the ORPO objective in (6) ."
>
>
> [1] Tomar, Manan, et al. "Mirror Descent Policy Optimization." International Conference on Learning Representations.
>
> [2] Grudzien, Jakub, Christian A. Schroeder De Witt, and Jakob Foerster. "Mirror learning: A unifying framework of policy optimisation." International Conference on Machine Learning. PMLR, 2022
>
> [3] Lu, Chris, et al. "Discovered policy optimisation." Advances in Neural Information Processing Systems 35 (2022).
>
> [4] Wang, Chaoqi, et al. "Beyond Reverse KL: Generalizing Direct Preference Optimization with Diverse Divergence Constraints." The Twelfth International Conference on Learning Representations.
>
> [5] Wu, Fan, and Patrick Rebeschini. "Implicit regularization in matrix sensing via mirror descent." Advances in Neural Information Processing Systems 34 (2021).
>
> [6] Ghai, Udaya, Elad Hazan, and Yoram Singer. "Exponentiated gradient meets gradient descent." Algorithmic learning theory. PMLR, 2020.
>
> Please let us know if you have any other questions or if there is anything else we could do to improve your assessment of our work!

---

> ### Comment · Reviewer_cDRz · 2024-11-24
>
> Thank you very much for your detailed feedback. Many of my questions were answered adequately and some concerns were addressed adequately. The code to reproduce the experiments was provided. So I will increase the score of reproducibility. The manuscript is also revised accordingly. Unfortunately, the revised parts are not highlighted in the manuscript or listed in the response. Therefore, it is hard to check how the manuscript was revised according to the reviewers’ comments. I strongly recommend to highlight the revised parts in the manuscript and provide a summary of changes in the response for your future submission.
>
> I still think that this paper can benefit from one more revision. Here are the suggestions.
>
> The overall framework of the proposed approach is still not clear. If I understand the response correctly, the agent can not access to the underlying MDP during the training. All it can access is the offline dataset. However, at the same time, the authors mentioned that the cumulative rewards were estimated online, meaning that it requires access to the underlying MDP. The estimation of the cumulative reward is required during the training phase to update the neural mirror maps. Therefore, it looks contradictory. Please correct me if it is wrong. If it requires both online and offline training, it will significantly limit the applicability of the proposed approach and providing example situations where the current setting is reasonable is required.
>
> For the analysis of the obtained loss function, first, I would like to thank you for addressing my comment. However, no evidence for the statement “this closed-form expression effectively captures the behavior and …” was given and the reason for a good performance, in particular compared to ORPO, is not discussed. To claim the transferability, I suggest the authors to perform deeper analysis on the obtained loss function.

---

> ### Author Response · Authors · 2024-11-25
>
> We thank the reviewer for their thoughtful feedback and for taking the time to reassess our work. Below, we address your comments in detail and provide clarifications where needed.
>
> **Summary of Changes**
>
> We recognize the importance of clearly documenting revisions for future submissions and appreciate your suggestion. In this case, since many sections were extensively rewritten, we refrained from highlighting changes directly in the manuscript to avoid confusion. Instead, we provide a summary of major revisions:
>
> - Abstract: Revised to reflect the updated contributions and findings.
> - Contributions (Introduction): The bullet points at the end of the Introduction were rewritten for clarity.
> - Section 2.2 (Mirror Maps): Expanded and clarified to address comments regarding the overall framework.
> - Experiments Section (Section 4): Completely rewritten to improve clarity and coherence.
> - Results Section (Section 5): Moved out of the Experiments section into its own dedicated section. This includes a formalization of the discovered objective in Subsection 5.6.
> - Conclusion (Section 7): Revised to better summarize the insights and implications of the study.
>
> **Clarification on Training Setup**
>
> Regarding your question about our training setup, we confirm that it follows a standard meta-learning framework, commonly employed in prior works on algorithm discovery [1, 2]. Here is a detailed explanation:
>
> During training, the agent only has access to offline trajectories, which it uses to optimize within the inner loop of our framework. This aligns with a meta-learning approach where the agent learns from limited data.
> Once the agent is trained on these offline trajectories, it is evaluated online—without further gradient updates—on a held-out set of states not observed during training. This evaluation phase simulates real-world testing conditions and measures the agent’s ability to generalize to unseen data.
> The resulting online evaluation scores are then used to select the best-performing loss functions, which are evolved in the outer loop of the optimization process. This iterative refinement ensures that the discovered loss functions lead to improved generalization.
>
> **Why Online Evaluation is Used**
>
> We opted for the MuJoCo environment specifically because it allows for cheap and accurate online evaluations, making it feasible to test the agent’s abilities in a practical setting. However, we emphasize that online evaluation is not a strict requirement of our approach. The framework could equally evaluate the agent on a held-out offline dataset not observed during training. While this would eliminate the need for online evaluations, it could introduce challenges in assessing the agent's generalization ability, as offline datasets may not fully capture the agent’s potential performance.
>
> This setting parallels the common scenario in LLM fine-tuning, where a model is trained offline on a preference dataset and later evaluated on a held-out set.
>
> **Addressing Loss Function Analysis**
>
> As stated in the paper, our loss function offers a smoother gradient compared to ORPO, which we believe contributes to more stable training, particularly in noisy environments. Additionally, the inclusion of a coefficient on the SFT term provides an advantage in scenarios where the dataset quality is inconsistent, such as when low-quality answers are occasionally labeled as "preferred" options.
> It is worth noting that most existing Preference Optimization [3] papers do not provide concrete explanations for the robustness or generalizability of their methods. Given the intricate dynamics of LLM training, it is inherently challenging to identify a specific factor driving these improvements beyond the explanations already outlined in our work.
> We find it noteworthy that our method has led to the discovery of such a simple loss function, which we believe serves as further evidence of its transferability.
> That said, we welcome any suggestions on how to conduct a more comprehensive analysis.
>
>
> [1] Lu, Chris, et al. "Discovered policy optimisation"
>
> [2] Lu, Chris, et al. "Discovering Preference Optimization Algorithms with and for Large Language Models"
>
> [3] Hong, Jiwoo et al. "ORPO: Monolithic Preference Optimization without Reference Model"

---

> ### Comment · Reviewer_cDRz · 2024-11-26
>
> Thank you for the clarification.
>
> Now, I believe my (initial) understanding of the training phase is correct. Every after the loss function update (outer-loop), the inner-loop optimization (offline policy training) must be performed. That is, to obtain the final policy, the proposed framework requires both an offline dataset and online interaction. Although the authors call the loss function update phase the evaluation phase, it is indeed used to train the policy (indirectly through the loss function update). Then, it is very misleading to say that the agent can only access the offline dataset. Moreover, because the related work section in this manuscript does not cover approaches utilizing online interaction during offline training, I have to say that the position of this paper is well-stated in the current manuscript.
>
> For the loss function analysis, as I mentioned in the previous comment, at least evidence for the statement “this closed-form expression effectively captures the behavior and …” should be given. One more thing is that the provided loss function is a (simplified approximation of a) loss function among several candidates obtained by the proposed approach. Therefore, just showcasing a single loss function is not sufficient to claim "We find it noteworthy that our method has led to the discovery of such a simple loss function, which we believe serves as further evidence of its transferability. "

---

> > ### Author Response · Authors · 2024-12-02
> >
> > We thank the reviewer for their feedback and appreciate the opportunity to clarify the concerns raised.
> >
> > **[Offline/Online training]**
> >
> > We would like to clarify the nature of our setting. First of all, we say that an agent or a model is trained online if new data is collected during training, for instance if the agent or model needs to be evaluated on new data to be updated. We say that an agent or model is trained offline if it is trained on data that was collected prior to training and no new data is collected during training, for instance if an LLM is fine-tuned on a preference dataset. Regardless of the training type, the trained agent is typically evaluated on new data, and therefore online. In our work, the MuJoCo agent (inner loop) is trained offline, while the PO objectives (outer loop) are trained online.
> >
> > - Inner Loop: the inner loop MuJoCo agent operates exclusively on an offline dataset and does not access the online environment at any point. This agent is trained with a candidate PO objective.
> > - Outer Loop Process: the outer loop, responsible for optimizing the PO objective, requires evaluation of the proposed PO objectives to perform an update and is therefore trained online. Its sole objective is to refine the loss function itself, ensuring it generalizes well to offline training scenarios.
> >
> > The distinction between these two steps is critical. The MuJoCo agent’s policy is indeed updated using the PO objective learned online by the outer loop, but the inner loop policy optimization process remains entirely within the offline domain. The objective of the meta-learning process (ES) is to generate an effective and transferable *offline* PO objective. We succeed at this, and demonstrate that the discovered objectives perform well on both MuJoCo and an LLM setting, where we only have access to a preference dataset.
> >
> > In practice, most offline PO algorithms in the literature have been validated by their empirical performance on online benchmarks like ChatBot Arena (add examples). The performance on these online benchmarks is what guides the progress and design of offline PO algorithms. Our methodology allows to validate the performance of PO algorithms in a systematic and efficient way, utilizing ES for the discovery of new PO algorithms, and therefore automating a process that, in other works, is manually performed by researchers. The only difference between our approach and the existing literature is that we replace the manual work done by an ML researcher with a meta-learning algorithm, but the fundamental methodology remains unchanged.
> >
> > We have updated the methodology section (lines 264-273) to further clarify our method, alongside newly added Appendix Sections (D, E and F) for further discussion and background information on offline and online algoritmns, meta-learning and evolution strategies. We hope these additions will be useful to better understand our method and how our work fits into the related literature.
> >
> > **[Simplified objectives]**
> >
> > We now provide two examples in the main text where the discovered objectives can be effectively summarized by a simple expression, namely the objectives discovered for the TLA environment in the noisy and shuffled dataset settings. Notably, both objective share the same structure, but with different coefficients, which are reported in the main text. We have further analysed the objective discovered on the base TLA dataset, which can be approximized by the expression in equation (11) of our paper with $\alpha=0.475$ and $\beta=0.630$, and the objective discovered on the Hopper dataset with noise 0.2, which can be approximized by the expression $$\pi^\star \in argmax_\pi E_{(s_0, \tau_w, \tau_l) \sim D} \left[1.557 \log(\pi(\tau_w)) +\lambda\log \sigma\left( f( \pi(\tau_w))-f(\pi(\tau_l))\right)\right]$$
> > where $f(x)=2.32x -13.37x^2-1.70x^3$. We report the value of the final policies trained using the simplified versions of the discovered objectives. We report the value in parenthesis: Hopper with noise 0.2 (1093 $\pm$ 105), base TLA (3467.5156 $\pm$ 67.986984), shuffled TLA (3344 $\pm$ 289), TLA with noise 0.1 (3752 $\pm$ 69). These values are slightly below the performance achieved by the neural-network-based objectives, proving the possibility of effectively approximating the discovered objectives with simple functions.
> >
> > To further show that the approximation is effective, we provide a plot for the original and simplified objectives discovered on base TLA at the anonymized link: https://imgur.com/a/na1qMJ2.

---

### Official Review · Reviewer_93E1 · 2024-10-30

**Soundness:** 3
**Presentation:** 2
**Contribution:** 2
**Rating:** 5
**Confidence:** 3

**Summary:**

This paper presents an empirical study on preference optimization (PO) algorithms, focusing on how preference dataset properties affect performance. The authors introduce a novel PO framework based on mirror descent, which generalizes existing methods like DPO and ORPO. A new framework for algorithm discovery, and the demonstration of generalization to LLM tasks have been proposed.

**Strengths:**

The paper proposes a novel framework for preference optimization (PO) based on mirror descent, which generalizes existing methods like DPO and ORPO. This approach of using mirror maps to explore the space of PO algorithms and discover new loss functions is relatively new in the field.
The idea of using evolutionary strategies to search for the best mirror maps and thus new PO algorithms allows for an automated and potentially more efficient way of finding algorithms that can handle different dataset properties compared to traditional methods of manually designing algorithms.
The application of the discovered PO algorithms from MuJoCo environments to LLM fine-tuning tasks shows the potential and generalization of algorithms discovered in a simpler, more computationally tractable environment to a more complex and practical domain like LLM tuning.

**Weaknesses:**

The discovered loss functions are complex and parametrized by neural networks. There is no attempt to provide an intuitive interpretation of what these functions represent or how they relate to the properties of the preference datasets. This lack of interpretability makes it difficult to understand and potentially modify the algorithms for specific applications.
The theoretical derivations, such as Theorem 3.1, are based on simplified assumptions. The impact of relaxing these assumptions or considering more complex scenarios where they may not hold is not explored.
The LLM fine-tuning experiment uses a modified dataset to simulate mixed-quality data, but it would be beneficial to validate the proposed algorithms on real-world, large-scale LLM datasets.
While the paper identifies certain failure modes of ORPO and shows how the proposed algorithms address them, the discussion could be more comprehensive. For example, exploring why these failure modes occur in the first place and what implications they have for the broader field of preference optimization could provide deeper insights.

**Questions:**

1) In the Hopper and Ant/TLA environments, why were the specific numbers of rows chosen for the datasets (5120 for Hopper and 1280 for TLA)? How do these choices impact the representativeness and generalizability of the results?
2) The evolutionary strategies used to search for the best mirror maps and loss functions are computationally expensive. Have you considered any techniques to speed up this process or approximate the optimal solutions more efficiently?
3) The discovered loss functions are complex and parametrized by neural networks. How can we ensure the stability and reliability of these functions in different application scenarios?
4) The generalization of the discovered PO algorithms from MuJoCo to LLM tasks is demonstrated, but it is not clear how the differences in the nature of the two domains (e.g., discrete vs. continuous actions, different data distributions) are accounted for. Can you provide more insights into this?

---

> ### Author Response · Authors · 2024-11-20
>
> We thank the reviewer for their insightful and valuable feedback. We have updated the paper to address the weaknesses raised.
>
> - We have included a section with a closed form version of the discovered loss function we transferred to the LLM setting since, as the reviewer correctly points out, it would be much easier to re-implement this simple one-line equation compared to loss function parametrized by a neural network.
> - Assumptions behind the theoretical derivation: we note that the assumptions behind Theorem 3.1 are: (1) the reference policy needs to be positive for each state action pair, i.e. $\pi_\text{ref}(a|s)>0$ for all $s\in\mathcal{S}, a\in\mathcal{A}$, and (2) the mirror map $h_\phi$ needs to be a 0-potential. The first assumption is always respected in LLM tuning tasks, where the probability distribution induced by the LLM is parametrized by a softmax head that assigns positive probability to all tokens. The second assumption can also be easily satisfied, as we can enforce any mirror map to be a 0-potential mirror map by adding the negative entropy mirror map. That is, for any mirror map $h$ and any constant $\alpha>0$, we have that $h'(x)= h(x)+\alpha\sum_a \log(x_a)$ is a 0-potential mirror map.
> - We have updated our discussion regarding ORPO and DPO failure cases and hope they are more comprehensive.
> - Although we agree additional experiments on a bigger, real-world, large-scale dataset would strengthen our findings, our computational  resources unfortunately make it impossible for us to provide results before the end of the rebuttal period, but we commit to reporting experiments before the camera ready deadline if the paper was to be accepted.
>
> > In the Hopper and Ant/TLA environments, why were the specific numbers of rows chosen for the datasets (5120 for Hopper and 1280 for TLA)? How do these choices impact the representativeness and generalizability of the results?
>
> The reason why ~5k was chosen is because it is equivalent to 10M total samples, which is the standard amount of samples needed to train a MuJoCo agent offline (1000 samples per trajectory and 2 trajectories for each row) [1].
>
> The reason why we only use ~1k samples in TLA is because we already start from a pretrained policy and all we require is a stylistic change, so less datapoints are needed.
>
> We have however tested our method on many different dataset sizes and we didn’t notice significant differences in the results so we don’t believe our choices affect the representativeness or generalisability of our results.
>
> > The evolutionary strategies used to search for the best mirror maps and loss functions are computationally expensive. Have you considered any techniques to speed up this process or approximate the optimal solutions more efficiently?
>
> While yes, ES is generally expensive, here we are carrying it out on environments that can be vectorised and parallelised using JAX, so it can be run efficiently on a single GPU. This allows for extremely fast training which has a fraction of the computational and time resources that are required to perform fine-tuning and RLHF even on a fairly small LLM.
>
> > The discovered loss functions are complex and parametrized by neural networks. How can we ensure the stability and reliability of these functions in different application scenarios?
>
> We thank the reviewer for their valuable feedback. We have updated the paper to include a closed-form equation which is equivalent to our discovered neural network loss.
>
> > The generalization of the discovered PO algorithms from MuJoCo to LLM tasks is demonstrated, but it is not clear how the differences in the nature of the two domains (e.g., discrete vs. continuous actions, different data distributions) are accounted for. Can you provide more insights into this?
>
> The different nature of action spaces (discrete vs continuous) was the main issue with transferring results from MuJoCo to LLM finetuning tasks. We addressed this challenge through an implementation trick, consisting in discretizing the policy of the MuJoCo agent during optimization. That is, the policy is parametrized by a neural network with a gaussian head as per MuJoCo standard procedure, but it is discretized before being inserted in the loss function and differentiated. This trick allows better transferability to LLM tuning tasks with discrete action spaces. We have modified the paper to include this discussion.
>
> [1] A Minimaximalist Approach to Reinforcement Learning from Human Feedback
>
> Please let us know if you have any other questions or if there is anything else we could do to improve your assessment of our work!

---

> > ### Comment · Reviewer_93E1 · 2024-11-26
> > **Thank you for the response**
> >
> > Thank you for the authors' response! Some of my concerns have been addressed, while there are still confusing parts. For example, ES is expensive not because the search operators but the evaluations. Function evaluations are problem-dependent, why this could be accelerated by GPU? This is not clear. So I would like to keep my score.

---

> ### Author Response · Authors · 2024-11-26
>
> Thank you for taking the time to read our reply.
>
> Regarding the computational cost of evaluations, we note that, as mentioned in Line 264 of our paper, we perform the evaluation step of ES in brax, a JAX compatible implementation of MuJoCo. In brax, we are able to run thousands of agents in parallel on a GPU and can therefore efficiently evaluate a generation of agents. In practice, the evaluation step of our procedure requires less than 2 minutes of computation on a single GPU (NVIDIA A40), which is order of magnitude smaller than the time, compute and memory it would require to evaluate or train an LLM.
>
> Please let us know if you have remaining concerns, otherwise we kindly ask that you consider raising our score.

---

### Official Review · Reviewer_t1gU · 2024-11-01

**Soundness:** 3
**Presentation:** 3
**Contribution:** 3
**Rating:** 6
**Confidence:** 3

**Summary:**

This paper proposed a new Preference Optimization (PO) framework .

Overall, I think the novelty of this paper is limited.

**Strengths:**

This paper provide a theoretical way to study preference optimization (PO) and give both theoretical and empirical results.

**Weaknesses:**

Experimental results are a bit too weak. Others see questions.

**Questions:**

1. Line 144. "A known issue of DPO is that it pushes probability mass away from the preference dataset and to
unseen responses". Is that true?

2. Line 133. Besides it computational costs, PPO is known to be prone to reward overoptimization. Is that true?

3. In which case DPO is better than RLHF, and which case RLHF is better than DPO. Is there an affirmative answer?

4. Line 240 to 242. Why these functions are considered?

5. As far as I understand, the proposed work try to change different Bregman distance. This is very similar to an ICLR work called "f-PO: Generalizing Preference Optimization with f-divergence Minimization". So, is there equivalence between each other? For example, by choosing a propose f divergence will yield a corresponding Bregman distance. I admit that choosing the Bregman distance is a clever idea.  However, what is the novelty becomes compared with FPO?

6. This paper mention "online " occurs a lot of times. What does it mean in preference optimization?

7. The experiments seems a bit weak. What is the performance on multi-modal model or 3B model?

**Details Of Ethics Concerns:**

NA.

---

> ### Author Response · Authors · 2024-11-20
>
> We thank the reviewers for their feedback and the opportunity to address the concerns raised.
>
> > Experimental results are a bit too weak
>
> - We show consistent and statistically significant improvements in performance against a range of identified failure cases for commonly used preference optimisation losses. These results are validated across multiple MuJoCo environments.
> - In addition to the MuJoCo experiments, we demonstrate the transfer of our discovered loss function to improve performance on a more complex large language model (LLM) task, specifically on a 7B-parameter model.
> - While we acknowledge the relevance of multi-modal LLMs in broader applications, our work is positioned within the well-established domain of fine-tuning text-only LLMs. None of the prior works in this area, including those on DPO and ORPO, include benchmarks for multi-modal LLMs. Therefore, we believe our focus is both appropriate and consistent with the existing literature.
> - We acknowledge that the paper could be strengthened by incorporating further experiments, particularly with a larger dataset. While we are unable to complete these additional experiments before the rebuttal deadline due to computational limitations, we are committed to including them into our work before the camera ready version.
>
> Replying now to the questions:
>
> > Line 144. "A known issue of DPO is that it pushes probability mass away from the preference dataset and to unseen responses". Is that true?
>
> Yes, we apologies for the missing reference on this sentence. We have updated the paper to include a reference to [1] where this phenomenon is explained in detail.
>
> >  Line 133. Besides it computational costs, PPO is known to be prone to reward overoptimization. Is that true?
>
> Yes. This is a well known event described in multiple works in the literature, we cite [4,5]. This is because the learnt reward model (represented by a neural network) may not generalise perfectly on all possible datapoints.
>
> > In which case DPO is better than RLHF, and which case RLHF is better than DPO. Is there an affirmative answer?
>
> Generally, Reinforcement Learning from Human Feedback (RLHF) is considered the superior approach, particularly when substantial amounts of data or online labels are accessible. This makes it the industry standard [1].
> While DPO has been theoretically equated to optimizing using Proximal Policy Optimization (PPO) and a reward model trained on an offline dataset, recent empirical research [3] has challenged this notion. These studies have demonstrated that online methods, such as PPO, consistently outperform offline methods like DPO. This superiority is attributed to the benefits of on-policy sampling.
>
> While DPO has occasionally outperformed PPO, it's important to note that several studies [1, 2] have consistently shown PPO's overall superiority. DPO's relative strength lies in its simpler training regime, which avoids the complexities associated with reward model inaccuracies. However, DPO's performance is significantly limited by its sensitivity to distribution shift, especially when the offline preference data lacks diversity [2]. This limitation becomes particularly evident when querying the model with out-of-distribution data, a common challenge for methods relying solely on offline data. To mitigate this issue, DPO-iter, which incorporates online data, has been proposed as a potential solution.
>
> > Line 240 to 242. Why these functions are considered?
>
> To ease the optimization procedure we are adopting a neural network with a relatively small number of parameters. To ensure a high representation power for the neural network, we are using monotonic activation functions with different curvature, i.e. with different second derivatives. A similar approach has been adopted by Lu et al. [6] and Jackson et al. [7].
>
> > As far as I understand, the proposed work try to change different Bregman distance. This is very similar to an ICLR work called "f-PO: Generalizing Preference Optimization with f-divergence Minimization". So, is there equivalence between each other? For example, by choosing a propose f divergence will yield a corresponding Bregman distance. I admit that choosing the Bregman distance is a clever idea. However, what is the novelty becomes compared with FPO?
>
> Thank you for pointing out this paper, we hadn’t seen it given it was published after the ICLR submission deadline. The only version of the paper “f-PO: Generalizing Preference Optimization with f-divergence Minimization" we could find is the one available on ArXiv, which was submitted two weeks ago or 1 month after our submission to ICLR. Please correct us if we found the wrong version. Regardless, the two frameworks are separate and the capability of recovering DPO is the only intersection. This can be observed by comparing their loss function (13) with our loss function in (9), as well as their expression for the optimal policy ((1) and (2) in their paper) with ours in (7) and (8).

---

> ### Author Response · Authors · 2024-11-20
>
> > This paper mention "online " occurs a lot of times. What does it mean in preference optimization?
>
> “Online” means we are able to gather new datapoints during training. “Offline” means we only have access to an given dataset and we cannot gather more datapoints. In the RLHF context, online means we would be able to query a human for each new generation of the LLM to give us an accurate judgment on which response is best. Offline means we start with a given preference dataset and we don’t have access to additional data. There are variants of DPO (which is traditionally offline) which incorporate online, on-policy data, like DPO-iter, and those perform much better [1]
>
> > The experiments seems a bit weak. What is the performance on multi-modal model or 3B model?
>
> As mentioned above, we tested on a bigger 7B model, so we believe the results on a 3B model would be similar.  Multi-modal LLMs are beyond the scope of this work, which places itself within the extensive literature on fine-tuning text-only LLMs. None of the works in this area include benchmarks multi-modal LLMs (DPO, ORPO etc).
>
> Please let us know if you have any other questions or if there is anything else we could do to improve your assessment of our work!
>
> [1] Xu et al. Is DPO Superior to PPO for LLM Alignment? A Comprehensive Study
>
> [2] Song et al. The Importance of Online Data: Understanding Preference Fine-tuning via Coverage
>
> [3] Tang et al. Understanding the performance gap between online and offline alignment algorithms
>
> [4] Coste et al. Reward Model Ensembles Help Mitigate Overoptimization
>
> [5] Gao et al. Scaling Laws for Reward Model Overoptimization
>
> [6] Lu et al. Discovered policy optimisation
>
> [7] Jackson et al. Discovering Temporally-Aware Reinforcement Learning Algorithms

---

> > ### Comment · Reviewer_t1gU · 2024-11-26
> > **nice work**
> >
> > nice work, and I have raised my score.

---

### Official Review · Reviewer_Vf5C · 2024-11-04

**Soundness:** 3
**Presentation:** 3
**Contribution:** 3
**Rating:** 6
**Confidence:** 3

**Summary:**

The paper investigates how preference optimization (PO) algorithms perform under varying conditions of data quality. Through empirical experiments primarily in MuJoCo environments, the authors examine the weaknesses of state-of-the-art PO methods like Direct Preference Optimization (DPO) and Odds-Ratio Preference Optimization (ORPO) when working with noisy or low-quality data. They propose a novel PO framework based on mirror descent to generalize existing PO methods and employ evolutionary strategies to identify optimal loss functions that could address identified performance issues. Results demonstrate that these new loss functions improve PO algorithm performance, even in fine-tuning tasks with large language models (LLMs).

**Strengths:**

1. The paper introduces a novel framework for loss optimization using mirror descent, which effectively addresses common issues in preference datasets.
2. The study includes well-designed experiments across various data settings.
3. The flexibility of the method across different application domains.

**Weaknesses:**

1. While the methodology is empirically validated, a deeper theoretical analysis, particularly around robustness and generalizability, would strengthen the findings.
2. The paper lacks a discussion on the computational cost of training with evolutionary strategies, especially for large models or in settings where data or compute resources may be constrained. The framework鈥檚 reliance on the computational complexity of evolutionary strategies might make it less practical.
3. The experimental settings focus primarily on MuJoCo, which, while diverse, may limit insights into real-world applications. Expanding beyond MuJoCo could improve the generalizability of results.

**Questions:**

1. How does the computational cost of this new framework compare with conventional PO methods in large-scale settings? Could efficiency trade-offs limit practical applications?
2. Are there specific criteria for choosing mirror maps or evolutionary strategies based on the characteristics of the preference dataset?
3. Could the approach be extended or adapted to reinforcement learning tasks beyond MuJoCo and language model fine-tuning? How would it perform in such settings?
4. More peer algorithms should be included for experimental comparison.

---

> ### Author Response · Authors · 2024-11-20
>
> We thank the reviewer for their kind words and interest in our work. Responding to the concerns raised:
>
> 1. The aim of our paper is to provide a framework for empirically discovering PO algorithms that are specialized to particular scenarios. We provide a theoretical justification for our family of algorithms, as we show in Theorem 3.1 that they are equivalent to maximizing the expected reward plus a Bregman divergence penalty. However, finding theoretical guarantees for the discovered algorithms or mathematically proving that our method can always discover efficient algorithms is beyond the scope of this work. We set these objectives as directions for future work. We also highlight that previous works on discovering algorithms, such as [1,2], do not contain theoretical results.
> 2. We have updated the paper to make this clearer, but the reason why all our evolutionary strategy optimisation is carried out on MuJoCo environments is exactly to make computation feasible. Thanks to environment and computation parallelisation using JAX, it is extremely computationally efficient to run ES on our problem (in wall clock time, it could take 1 or 2 days to optimise a loss function on a single, small GPU). As comparison, this much cheaper (both in terms of time and memory requirements) than running even a few epochs of finetuning on a medium size (7B parameters) LLM.
> 3. This is why we demonstrate that our discovered loss functions are general enough to transfer to an LLM finetuning setting. Moreover, RLHF is a method that was originally developed for robotics applications (so very similar to our MuJoCo setting) showing our method is widely applicable across a wide range of settings.
>
> Replying to the questions:
> 1. Our method is much, much cheaper than running any experiments on an actual LLM, and allows practitioners to gain intuition and a prior around which loss functions could be best suited for their LLM task without having to run very expensive experiments on an LLM. This is particularly relevant to researchers with very constrained resources.
> 2. Any evolutionary strategy could be used for our framework, but we choose OpenES due to its proven track record on similar problem settings [1,2].
> 3. The experiments on an LLM finetuning task are included in the paper. On a highlighted ORPO failure case, we outperform ORPO according to the Alpaca Eval 2 evalutation.
> 4. We absolutely agree including more baselines would strengthen the paper. DPO was left out as it was being outperformed by ORPO (we updated the paper to include more discussion regarding this point). We are working on adding additional baselines right now, and will update the paper as soon as they are ready. We hope this will be useful to inform practitioners on the strengths and weaknesses of each loss function depending on the setting they are applied to.
>
> [1] Lu et al. Discovered policy optimisation
> [2] Jackson et al. Discovering Temporally-Aware Reinforcement Learning Algorithms
>
> Please let us know if you have any other questions or if there is anything else we could do to improve your assessment of our work!

---

> ### Comment · Reviewer_Vf5C · 2024-11-27
>
> Thank you for the authors' detailed response! I have no further questions.  I would like to maintain my score.

---

### Meta-Review · Area_Chair_K2La · 2024-12-19

**Metareview:**

This paper introduces Mirror Preference Optimization (MPO), a framework that generalizes existing preference optimization methods, including Direct Preference Optimization (DPO) and Odds-Ratio Preference Optimization (ORPO) via mirror descent. The authors use evolutionary strategies to discover specialized loss functions tailored to noisy or mixed-quality datasets. Empirical evaluations in MuJoCo environments show improvements over baselines, and the discovered loss functions transfer successfully to large language model fine-tuning tasks.

**Strengths**
* MPO offers a principled generalization of existing PO methods
* The experiments in MuJoCo show consistent improvements over baselines, and the transfer to LLM tasks is promising.
* The use of ES for discovering loss functions helps automate algorithm design.

**Weaknesses**
* Limited comparisons with specialized DPO variants (e.g., cDPO, rDPO) hinder a comprehensive assessment of novelty.
* Lacks robust theoretical guarantees on generalizability.
* Validation is focused on MuJoCo with limited real-world testing.

The paper presents an interesting and potentially impactful approach, but the weaknesses — particularly the lack of baseline comparisons, theoretical analysis, and broader experimental validation — reduce its overall significance. While the approach shows promise, these issues currently outweigh the paper's strengths.

**Additional Comments On Reviewer Discussion:**

During the discussion phase, reviewers recognized the potential of the MPO framework but maintained concerns. While Reviewer cDRz raised their score slightly, noting the novelty of applying ES to preference optimization, they emphasized the need for stronger comparative analysis and theoretical guarantees. Furthermore, there are unresolved issues with baseline comparisons, methodological rigor, and broader validation. To improve, the authors are encouraged to include comparisons with specialized DPO variants (e.g., cDPO, rDPO), provide deeper theoretical analysis, and expand experiments to real-world scenarios beyond MuJoCo and LLM tasks. Clarifying the framework’s technical advantages over existing ES-based methods would also strengthen its contribution.

---

### Decision · Program_Chairs · 2025-01-22

Reject